# Com1 as a Promising Protein for the Differential Diagnosis of the Two Forms of Q Fever

**DOI:** 10.3390/pathogens8040242

**Published:** 2019-11-18

**Authors:** Iosif Vranakis, Eirini Mathioudaki, Sofia Kokkini, Anna Psaroulaki

**Affiliations:** 1Laboratory of Clinical Microbiology and Microbial Pathogenesis, Unit of Zoonoses, School of Medicine, University of Crete, 71110 Heraklion, Greece; iosif.vranakis@gmail.com (I.V.); chemp927@edu.chemistry.uoc.gr (E.M.); sofkok@hotmail.com (S.K.); 2Laboratory of Biochemistry, School of Science and Engineering, Department of Chemistry, University of Crete, 71110 Heraklion, Greece; 3Intensive Care Unit, General Hospital of Agios Nikolaos, 72100 Agios Nikolaos, Greece

**Keywords:** Com1, *Coxiella burnetii*, antigenic, Q fever, ELISA

## Abstract

*Coxiella burnetii* is the causative agent of acute and chronic Q fever in humans. Although the isolates studied so far showed a difference in virulence potential between those causing the two forms of the disease, implying a difference in their proteomic profile, the methods used so far to diagnose the two forms of the disease do not provide sufficient discriminatory capability, and human infections may be often misdiagnosed. The aim of the current study was to identify the outer membrane Com1 (CBU_1910) as a candidate protein for serodiagnostics of Q fever. The protein was cloned, expressed, purified, and used as an antigen in ELISA. The protein was then used for the screening of sera from patients suffering from chronic Q fever endocarditis, patients whose samples were negative for phase I immunoglobulin G (IgG), patients for whom at least one sample was positive for phase I IgG, and patients suffering from any kind of rheumatoid disease. Blood donors were used as the control group. Following statistical analysis, 92.4% (122/132) of the samples tested agreed with the negative clinical diagnosis, and 72.2% (26/36) agreed with the positive clinical diagnosis. Moreover, a significant correlation to the presence of the disease (*p* = 0.00) was calculated. The results support the idea that a Com1 antigen-based serodiagnostic test may be useful for differential diagnosis of chronic Q fever. Further studies are required to compare more immunogenic proteins of the bacterium against samples originating from patients suffering from different forms of the disease.

## 1. Introduction 

Q fever is a zoonosis caused by the, until recently obligatory, intracellular bacterium *Coxiella burnetii* [1]. The disease was largely considered as an occupational one since humans need to come into contact with an infected animal to get infected. Even though a vast majority of mammals can act as reservoirs of the bacterium, sheep, goats, and cattle are the primary animal reservoirs. This is the reason why people at higher risk for *C. burnetii* infection include vets, slaughterhouse workers, farmers, and people in general who come into contact with animals of veterinary importance. However, the fact that the pathogen is primarily spread by contaminated aerosols and can travel via wind to large distances due to its small size, as well as its remarkable viability against environmental conditions and its extremely low infection dose, the originally strong belief that Q fever is a strictly occupational disease is starting to decline. As proof of this statement comes the Netherlands Q fever outbreak in 2009 during which more than 4000 Q fever cases were reported [2]. During the past 25 years, the 32 outbreaks identified in Europe indicate that the number of Q fever cases is increasing [3,4]. Inevitably, the increase of the socioeconomic burden follows from the infection that presents significant challenges for both public and animal health [5].

As seen in bacteria of the Enterobacteriaceae family, *C. burnetii* displays antigenic variations. Phase variation of the pathogen is related to the mutational variation in its lipopolysaccharide (LPS) [6]. Non-infectious phase II bacteria corresponding to rough LPS are obtained in laboratories following serial passages in cell cultures. Bacteria in phase I (natural phase corresponding to smooth LPS) are detected in humans and animals. Bacteria in phase I are highly infectious [6]. 

In small ruminants *C. burnetii* infection presents mostly without clinical symptoms; nevertheless, abortions and stillbirths can occur mainly during late pregnancy and can lead to high economic burden. Shedding of the pathogen occurs mostly in placental membranes and birth fluids during parturition of infected small ruminants; therefore, birth products act as source of bacteria that become aerosolized and transmitted to humans. In humans, the disease may present with various acute and chronic clinical manifestations [7]. The incubation period before the onset of symptoms can last from two to three weeks depending on the size of the inoculum. Acute infection can present through a wide diversity of clinical symptoms, while, in a large proportion of patients, infection may be asymptomatic [8]. In other cases, pneumonia, hepatitis, or flu-like syndrome were described.

A small proportion of the patients infected by *C. burnetii* progress to chronic Q fever, with endocarditis being the main clinical manifestation [6]. Chronic Q fever leads to high death rates if left untreated, which makes early diagnosis and proper antibiotic administration critical for patients at high risk.

Since these extremely polymorphic clinical symptoms of the infection cannot be diagnostic for Q fever, diagnosis is largely based on laboratory diagnostic tools. *Coxiella burnetii* cannot be cultivated using standard routine laboratory culture techniques; therefore, laboratory diagnosis is based on indirect diagnostic tools. Antibody detection is the most common method for testing for *C. burnetii* infection. Indirect immunofluorescence assay (IFA) is the reference method, but the complement fixation test (CFT) and ELISA are also used. In IFA, an immunoglobulin G (IgG) anti-phase II antibody titer of ≥ 200 and an IgM anti-phase II antibody titer of ≥ 50 are generally considered significant for the laboratory diagnosis of the acute phase of infection [6] Chronic Q fever is characterized by the presence of anti-phase I antibodies, and an IgG anti-phase I antibody titer of ≥ 1/800 is generally considered to be highly predictive of Q fever endocarditis. In any case, since there is a large variability from one area to another, cut-off titers also differ; therefore, their choice depends on the prevalence of antibodies against the pathogen in the population under study [3,6].

However, IFA appears to have several disadvantages, such as the requirement of acute and convalescent sera, the objectivity of the interpretation of the results, potential antibody cross-reactions (false positive samples), the need for experienced personnel, etc. On the other hand, the advantage of ELISA is that it is easy to perform, interpretation is less subjective compared to IFA, and automation is possible. However, commercially available Q fever ELISA tests lack the ability of IFA to identify patients at risk for developing chronic Q fever [9].

In this context, we identified and determined antigenic proteins which could be used for the development of a chronic Q fever-specific ELISA and finally evolve into a fast immunochromatographic kit. A strong candidate protein that could fulfill these prerequisites is the outer membrane protein Com1.

The aim of this current study was to test the ability of CBU_1910 (Com1), an outer membrane protein from *C. burnetii*, to differentially diagnose chronic Q fever in humans.

## 2. Materials and Methods

### 2.1. Bacterial Strains, Plasmids, Media and Growth Conditions

In this study we used the *Escherichia coli* C43 (DE3) strain and LB (Luria–Bertani) medium. The *E. coli* cells were cultured aerobically at 37 °C. Carbenicillin was used in the culture medium or plates at the concentration of 50 mg/L. The plasmid used was pET-22b-CBU_1910, which contains the *lac* operon, the structural gene encoding CBU_1910, and a His tag in the C-terminal.

### 2.2. Cloning of CBU_1910

For the construction of the plasmid pET-22b, genomic DNA from *C*. *burnetii*, strain RSA 493/Nine Mile phase I was used. The cloning procedure was previously described by Sekeyova et al. [10]. In this study, the PCR primers were designed according to the CBU_1910 sequence from GenBank National Center for Biotechnology Information (NCBI). PCR was performed (Phusion NEB M0530) with gene-specific primers CBU_1910_Forward (F) (5′–GCCCATATGAAGAACCGTTTGACTGCACTATTTTTAGCC–3′) and CBU_1910_Reverse (Rev) (5′–GTCCTCGAGCTTTTCTACCCGGTCGATTTCTTTTTGAAG–3′). The restriction endonuclease *Nde*I and *Xho*I sites were added in the forward and reverse primers, respectively, and they were used at final concentration of 10 pmol/μL to amplify a 750-bp segment.

Amplification consisted of an initial 3-min step at 94 °C followed by 30 cycles of 40 s at 94 °C, 30 s at 56 °C, and 5 min at 72 °C. The PCR product and pET-22b vector were digested by *Nde*I and *Xho*I and then ligated using DNA ligase. The plasmid was then transformed in DH5α *E. coli* cells. Plasmid DNA from those contracts was analyzed by DNA sequencing (Eurofins MWG Operon (Ebersberg, Germany). The plasmid was transformed into DH5α *E. coli* cells. This construct was designated as pET-22b-CBU_1910. 

### 2.3. Expression and Purification of CBU_1910

Expression of CBU_1910 was tested in several conditions, such as different culture temperatures, different Isopropyl β- d-1-thiogalactopyranoside (IPTG) concentration for cell induction, and different time points of removing cell samples, after induction.

Expression of CBU_1910 was performed by transforming the plasmid into competent *E. coli* cells. The amount of 25 mL of an overnight culture of *E. coli* cells (pET-22b- CBU_1910) was inoculated into 1 L of LB broth containing carbenicillin (50 mg/L) and incubated at 37 °C until an Optical Density OD_600_ of 0.5 was reached. Induction was achieved by adding IPTG in a final concentration of 1 mM. Four hours after induction, the cells were collected by centrifugation at 4300× *g* for 10 min in an HFA 14,290 Heraeus SepatechSuprafuge 22 rotor.

Purification of the recombinant CBU_1910 was achieved by using a 4-mL Ni- nitrilotriacetic acid (NTA) indole butyric acid affinity column (IBA), as described by Sekeyova et al. [10] with some modifications. Briefly, the cell pellet was resuspended in 20 mM hydroxyethyl piperazineethanesulfonic acid (HEPES), 300 mM NaCl, 1 mM Ethylenediaminetetraacetic acid (EDTA) and 1 mM phenylmethylsulfonyl fluoride (PMSF) (pH 7.5) and incubated on ice for 20 min. Cells were lysed using a French Press (SLM Aminco, Model: FA-078). Lysed cells were centrifuged at 17,400× *g*, 4 °C for 20 min. The supernatant was ultra-centrifuged for 1 h at 200,000× *g* in a 70 Ti Sorvall rotor. The pellet was collected and solubilized using *n*-dodecyl β-d-maltoside (DDM) in a final concentration of 1% for 1 h at 4 °C. After ultracentrifugation for 1 h, 200,000× *g* at 4 °C in a 70 Ti Beckmann rotor, the supernatant was collected.

The supernatant was loaded onto the Ni-NTA affinity column, which was equilibrated with 20 mM HEPES, 300 mM NaCl, and 1 mM EDTA (pH 8.0) and 0.1% DDM. The column was initially washed with 10 column volumes of 10 mM imidazole and then with 10 column volumes of 30 mM imidazole. The His-tagged recombinant protein was eluted by five column volumes of 300 mM imidazole. The eluted protein was collected and concentrated using a 30-kDa cut-off amicon in a final volume of 1000 μL.

### 2.4. Detection of CBU_1910 by Immunoblotting 

The amount of 20 μg of the purified recombinant CBU_1910 was suspended in sample buffer (12% SDS, 10% glycerol, 6% mercaptoethanol, 0.05% Serva Blue G, and 150 mM Tris-HCl, pH = 7), incubated for 10 min at 37 °C, separated on a 12% SDS-PAGE gel, and transferred onto nitrocellulose membranes (PerkinElimer Life Sciences). Membranes were then blocked for 1 h with 2% albumin biotin in tris-buffered saline (TBS) and Tween 20 (TBST) (150 mM NaCl, 0.05% Tween-20, 10 mM Tris/HCl, pH = 8). Blots were incubated for 1 h with anti-polyhistidine-alkaline phosphate antibody (Sigma) at a 1:2000 dilution. Finally, the blots were developed using a 5-bromo-4-chloro-3-indolyl phosphate/nitro blue tetrazolium (BCIP/NBT) system. The BCIP molecule was hydrolyzed by Alkaline Phospatase (AP) buffer (100 mM NaCl, 5 mM MgCl_2_, and 100 mM Tris/HCl, pH = 9.5), and the produced molecule dimerized to give a colorful product, including the protein of interest. In addition, we used sera from patients diagnosed with acute or chronic Q fever (according to laboratory IFA test results and clinical diagnosis; see Section 3.4), in order to test the CBU_1910’s ability to correctly identify the chronic Q fever sample.

### 2.5. Serum Samples

Sample sera used for this study were borrowed from the sera collection of the Hellenic national reference center for Q fever which is located in Crete, in the southern part of Greece. All patients were asked for their informed consent concerning future research studies utilizing their samples (questionnaire form). Sample sera were initially tested with immunofluorescence (IFA) for total immunoglobulin directed against phase II *C. burnetii* for IgG and IgM antibodies, using a commercially available antigen (*C. burnetii* spot IF; Bio-Merieux, Marcy L’Ιtoile, France). All sera corresponding to patients under the suspicion of chronic Q fever (endocarditis, hepatitis, etc.) were also tested for phase I IgG antibodies, using a commercially available IFA kit (FOCUS Diagnostics, Cypress, CA, USA). All sample sera chosen for analysis using the under-development ELISA were re-tested using the FOCUS diagnostic kit.

### 2.6. ELISA Development and Optimization

Titrations were performed via an indirect ELISA method to identify the optimal antigen and conjugate concentration as described elsewhere [11]. Microtiter flat-bottom plates (96 wells, Corning Costar, USA) were coated with different concentrations of CBU_1910 (1.0, 0.5, 0.25, 0.125, 0.0625, 0.03, and 0.015 µg∙mL^−1^) diluted in carbonate–bicarbonate buffer (pH 9.6) and incubated overnight at 4 °C. The wells were then washed and blocked with 1% bovine serum albumin (BSA) in Phosphate Buffer Saline (PBS) with 0.05% Tween-20 (washing buffer) for 60 min at 37 °C. The wells were thereafter incubated with a solution of Q fever-positive serum obtained from the library of the laboratory (Section 3.4) serially diluted (1:25 to 1:6400) in washing buffer + 1% BSA (blocking buffer) for 60 min at 37 °C; then, they were washed three times with washing buffer and were incubated with rabbit anti-human IgG (Boster Biological Technology) with peroxidase as a conjugate that was diluted 1:3000 in blocking buffer for 1 h at 37 °C. Next, the wells were washed three times and were further incubated with a solution of substrate/chromogen (TMB Core, BioRad, California, USA) for 10 min. The colorimetric reaction was stopped by adding 0.5 M H_2_SO_4_. A volume of 100 µL of the reagent mixture was added to each well. The optical density was read at 490 nm on a microplate reader (Multiskan FC, Thermo Scientific, Ratastie Finland).

The optimal concentration of the antigen was determined by the lowest concentration which demonstrated the positivity of the reaction at any dilution of the positive serum. The optimal dilution of the serum was presented by the greatest difference in reading between the positive and negative serum in the optimal antigen concentration.

### 2.7. Statistical Analysis

All statistical analysis was conducted using the IBM SPSS Statistics Version 25 statistical package. A *p*-value < 0.05 was considered as statistically significant. A receiver operating characteristic (ROC) curve analysis was used; following the area under curve (AUC) estimation, the cut-off value maximizing the Youden’s index (J = sensitivity + specificity − 1) was selected as the optimal cut-off value. The protein was considered as positive for the new diagnostic test (ELISA) following its classification to the two categories of 0 = “no disease” or 1 = “disease”, by considering each value lower than the cut-off as 0 (no disease) and each value greater than or equal to the cut-off as 1 (disease). Then, the corresponding classification 2 × 2 table of CBU_1910 was obtained in reference to the standard clinical classification. The values of true positives, true negatives, false positives, and false negatives of that table were used for the calculation of a series of indicators of the agreement between the classification obtained by the protein and the existing clinical classification, i.e., indicators of the performance of the protein as an antigen in a possible diagnostic test: sensitivity, specificity, positive predictive value, negative predictive value, and Cohen’s kappa coefficient (κ).

## 3. Results 

### 3.1. Cloning of CBU_1910

An expression vector pET-22b was constructed in order to express the CBU_1910 protein, a 27.6-kDa outer membrane protein, in the *E. coli* C43 (DE3) strain. The promoter and the transcription terminator regions of the *lac* operon, as well as the structural gene encoding CBU_1910, were amplified by PCR. An isolating tag (His-tag) was also fused to the C-terminal of the structure gene for the isolation of the gene translation product. The full sequence of this construct can be found in the Appendix A.

### 3.2. Expression and Purification of CBU_1910

CBU_1910 was best expressed at 37 °C, induction was performed with the use of 1 mM IPTG, and cell collection was carried out 4 h after induction. Figure 1b shows the corresponding Western blot result for Com1, a 27.6-kDa outer membrane protein, using the anti-His antibody. As a control, we used the whole-cell lysate just before induction (OD_600_ = 0.5, t = 0).

The protein was isolated and purified using a nickel affinity column. Figure 1c shows the SDS-PAGE Coomassie-stained gel of the pure protein.

### 3.3. Immunotesting of Acute and Chronic Q Fever 

Sera from patients diagnosed with acute or chronic Q fever were used, in order to test the CBU_1910’s ability to correctly identify chronic Q fever. In our study, we used sera from three different patients suffering from chronic Q fever and three different patients suffering from the acute form of the disease. Figure 2 displays a representative immunoblotting result. As a control, we used the whole-cell lysate before induction (t = 0), where our protein of interest was absent.

### 3.4. Sample Sera

A total of 160 sample sera were re-tested using the IFA technique (FOCUS Diagnostics, Cypress, CA, USA). Out of these 160 samples, 20 samples originated from chronic patients (based both on laboratory (≥1/1024 IgG phase I) and clinical findings). The lowest observed IgG phase I titer in this group was 1/4096. The blood donor group, considered as our healthy population Q fever-negative sample, consisted of 12 sample sera. All but one serum indicated titers lower than 1/64 against IgG phase I. Interestingly, one blood donor sample indicated a titer of 1/4096 positive for chronic Q fever according to IFA results (IgG phase I ≥ 1/1024). Since the sample serum originated from a blood donor (not a patient), we believe that this was a case of a false positive test result by IFA. On the other hand, acute Q fever may be asymptomatic. In this case, the observed high-titer antibodies are directed against phase I of the bacterium (indicative of chronic Q fever), which cannot be the case in a healthy individual. Unfortunately, although it would be of particular interest to follow up this individual’s serological profile in order to see whether we are talking about a false chronic Q fever-positive result or an asymptomatic acute Q fever case, we failed to get access to a second sample. Patients with high immunoglobulin titers against *C. burnetii* IgG phase II but lower than 1024 against phase I constituted our next group and consisted of 61 samples. The next group of patients consisted of 47 sample sera which indicated at least one sample ≥ 1/1024 IgG phase I (positive laboratory diagnosis). However, the final diagnosis for this group of patients was not chronic Q fever. Finally, 20 sample sera from patients with rheumatoid disease constituted our last group of samples. As commonly known, samples originating from patients with any kind of rheumatoid disease can cross-react in immunofluorescence assays, providing false positive results; three out of 20 samples indicated IgG phase I titers higher than 1/2048.

Table 1 summarizes the groups of patients’ sample sera that were re-tested with IFA and later analyzed with the indirect ELISA under development. 

### 3.5. Indirect ELISA

The titration of CBU_1910 and Q fever-positive serum determined by indirect ELISA showed high levels of reactivity in nearly all combinations of concentrations tested. In view of this result, the ideal protein antigen to be applied in the indirect ELISA with the highest reactivity combined with the minimum “background” interference was defined at a protein concentration of 1 μg/L and a serum dilution of 1/100.

### 3.6. Statistical Analysis

The patients tested were initially divided into five groups: (a) chronic patients (based both on laboratory and clinical examination), (b) blood donors, (c) patients whose samples were negative for phase I IgG, (d) patients for whom at least one sample was positive for phase I IgG, and (e) patients suffering from any kind of rheumatoid diseases. During the second step of analysis, samples were divided into two categories: those presenting an IgG phase I titer of ≥1/1024 and those below this cut-off point.

The cut-off point for CBU_1910 was calculated based on this laboratory cut-off and on the distribution of the samples which corresponded to true positive chronic patients (Table 2). Based on the protein’s cut-off, we calculated the discrepancy between the results obtained following the final diagnosis set up by the clinicians against that of the CBU_1910 ELISA (Table 3). 

According to the statistical analysis, 92.4% (122/132) of the samples tested agreed with the negative clinical diagnosis, and 72.2% (26/36) of the samples tested agreed with the positive clinical diagnosis. 

Following the ROC curve analysis, the true and false negative and the true and false positive values were calculated for CBU_1910. Furthermore, we calculated a significant correlation to the presence of the disease (*p* = 0.00), while the area under the curve was 0.974 (Table 4, Figure 3).

## 4. Discussion 

The interest of this study lies in the search for a reliable serological diagnostic tool for the differential diagnosis of Q fever. CBU_1910 (Com1) protein was chosen as it is known to be preferentially exposed on the surface of *C. burnetii* [12] and is generally considered as a credible Q fever serodiagnostic marker.

This protein is targeted during the early humoral immune response in acutely infected guinea pigs of the Nine Mile strain in phase I [13] and in vaccinated cattle [14]. In a study by Vigil et al. (2011) [15], during their attempts to profile the humoral immune response of the acute and chronic forms of the disease by protein microarray, the authors concluded that Com1 showed significantly less sero-reactivity between the acute and the chronic phase of infection, compared to other proteins [15]. Moreover, assays that used Com1 and were based on ELISA were able to distinguish vaccinated cattle from naturally exposed ones [13]. Moreover, Com1-vaccinated humanized mice [human leukocyte antigen (HLA)-DR4 transgenic) were shown to induce a strong gamma interferon recall response in purified CD4^+^ (cluster of differentiation 4) T cells [16]. In fact, Beare et al. [17] utilized this protein in a diagnostic ELISA and recorded higher specificity compared to the *C. burnetii* cell extract (specificity/sensitivity of 90.0/50.0 compared to 87.5/85.0, respectively). Zhang and coworkers utilized Com1 and proposed that an ELISA with the 27-kDa recombinant antigen is sensitive and specific enough to detect antibodies against the pathogen in human sera [18]. In an additional study carried out by Chen et al. (2014) [19], the authors evaluated the ability of the protein to detect the pathogen’s antibodies in an ELISA with peroxidase-based signal amplification [19].

ELISA based on *Com1* was also used to screen sera from patients suffering either from Q fever endocarditis or from the acute phase of the disease [10]. During that study, the authors calculated the values of specificity and sensitivity at the levels of 90% and 50%, respectively. These findings allowed the authors to suggest that an ELISA based on Com1 would be better applicable for the diagnosis of Q fever endocarditis than for the acute phase of the disease.

In this context, we used Com1 for its ability to detect antibodies specific for the chronic form of the disease. The results described above show that this protein is indeed immunologically responsive to sera from patients with chronic Q fever, while showing no response against sera from patients suffering from the acute form of the disease. The subsequent statistical analysis showed that CBU_1910 has sufficient specificity (92.4%) and sensitivity (92.9%) against the chronic form of the disease; thus, it can be considered as an antigen for the development of a chronic Q fever-specific diagnostic tool. The increased sensitivity and specificity observed in this study compared to previous studies [10,17] is probably due to two reasons. Firstly, our study focused on the chronic form of Q fever and the ability of the protein to adequately diagnose this particular form of the disease. Secondly, the “local” *C. burnetii* strain may be different enough to cause the Greek Q fever patients’ immune system to respond differently. Nevertheless, the fact that the Com1 ELISA identified 35 sample sera with antibody titer ≥ 1024 according to IFA as chronic Q fever-negative samples (Table 1; 32 samples with acute Q fever, three samples with rheumatoid disease) is an encouraging result.

Apparently, in order to increase the sensitivity and specificity of the under-development diagnostic assay, additional chronic Q fever-specific antigenic proteins should be studied and implemented into the assay. Using a pool of chronic Q fever-specific proteins instead of one combined with acute Q fever-specific proteins should hypothetically increase the qualitative characteristics of the assay. Finally, a larger number of sample sera should be analyzed to eliminate any potential bias.

## Figures and Tables

**Figure 1 pathogens-08-00242-f001:**
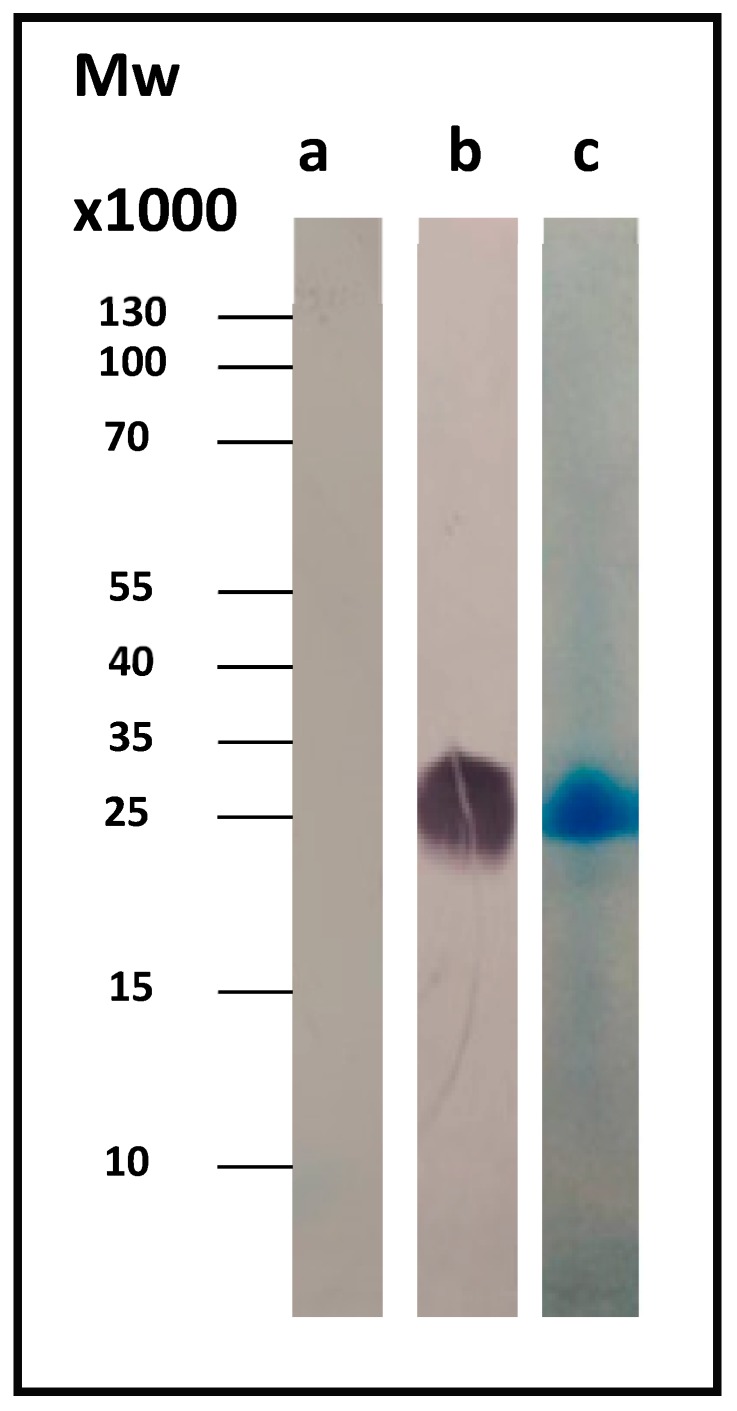
(**a**) *Escherichia*
*coli cells* before induction (t = 0). (**b**) Immunoblot for the expression the CBU_1910 gene in *E. coli* cells (t = 4 h after induction). (**c**) SDS-PAGE for the isolated and purified protein CBU_1910, a 27.6-kDa outer membrane protein.

**Figure 2 pathogens-08-00242-f002:**
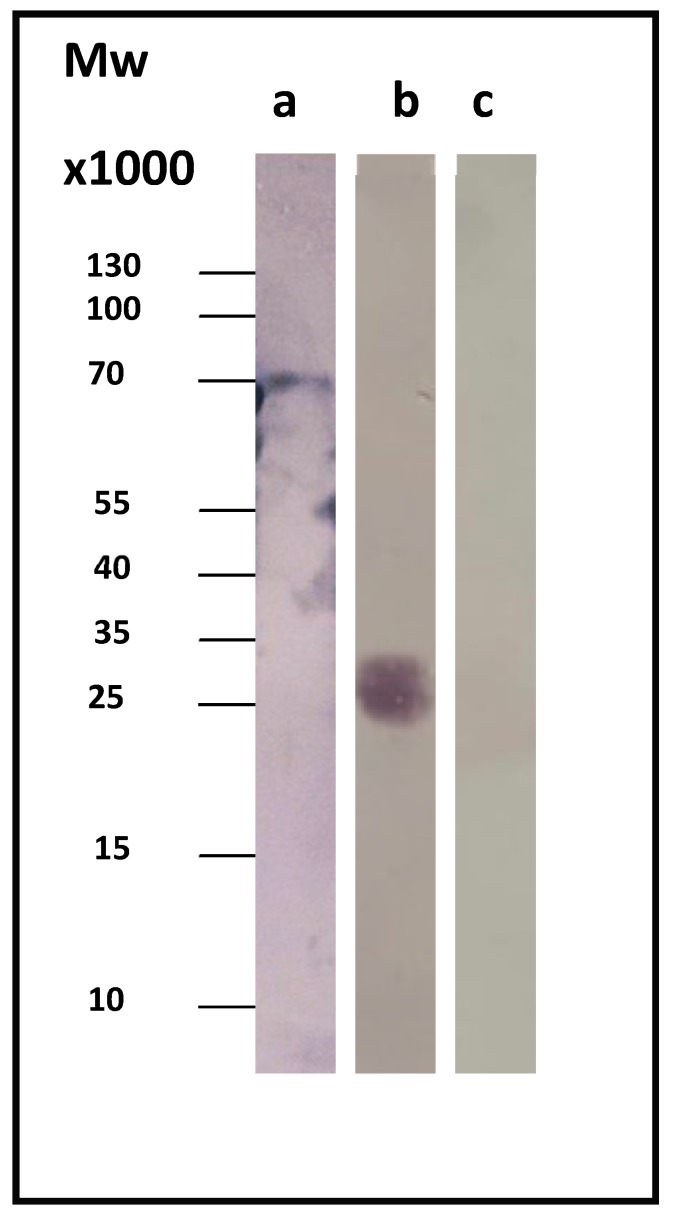
(**a**) Control: whole-cell lysate before induction. (**b**) Immunoblotting using serum from a patient suffering from chronic Q fever (immunofluorescence assay (IFA) result: immunoglobulin G (IgG) > 4096 and endocarditis). (**c**) Immunoblotting using serum from a patient suffering from acute Q fever.

**Figure 3 pathogens-08-00242-f003:**
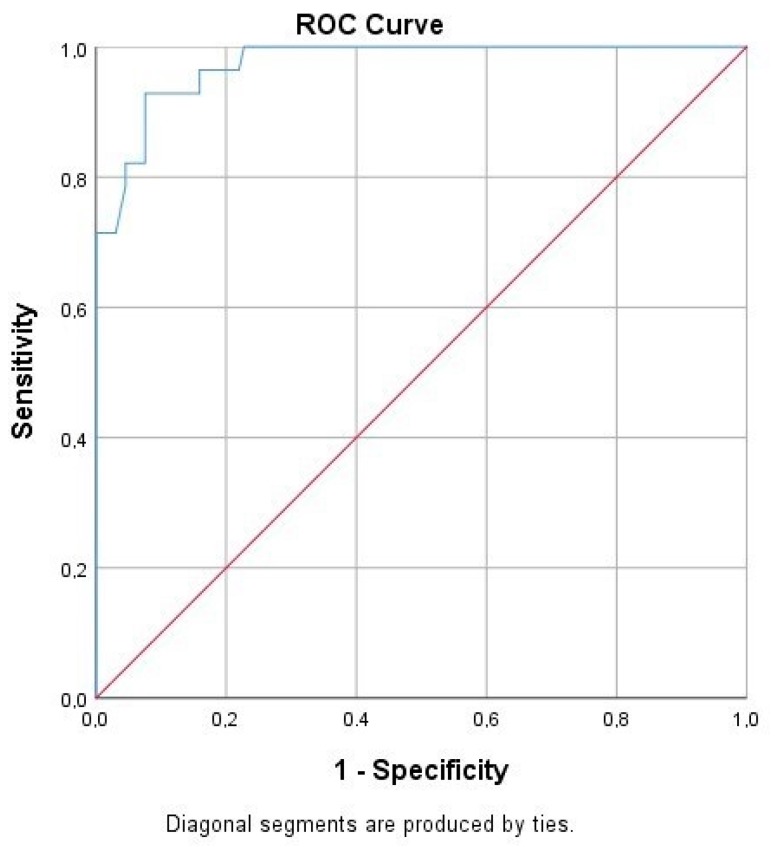
Sensitivity and specificity of the protein CBU_1910 based on the receiver operating characteristic (ROC) curve analysis.

**Table 1 pathogens-08-00242-t001:** Sample sera categorized according to the Q fever laboratory diagnosis and the final diagnosis (laboratory diagnosis and clinical findings). IgG—immunoglobulin.

Number of Sample Sera	Clinical Diagnosis	Laboratory Diagnosis	
		Positive for Acute Q Fever ≥1/1920 IgG Total	Positive for Chronic Q Fever ≥1/1024 IgG Phase I
20	Chronic Q fever		100%
12	Blood donors		8.3% (1/12)
61	Acute Q fever	100%	0
47	Acute Q fever (at least one sample of each patient was tested ≥1/1024 IgG phase I)	100%	68.1% (32/47)
20	Rheumatoid disease	55% (11/20)	15% (3/20)

**Table 2 pathogens-08-00242-t002:** True and false negative (tn and fn), and true and false positive (tp and fp) values as calculated based on the receiver operating characteristic (ROC) curve analysis for CBU_1910. From the table, the number of samples agreeing with the clinical diagnosis and presenting with proteins above or below the cut-off set up, for each group in the current study, can be extracted.

Protein	Clinical Diagnosis	ELISA		Total (%)
		Negative (%)	Positive (%)	
1910	Negative	122	10	132 (82.5%)
	Positive	2	26	28 (17.5%)
	Total	124 (77.5%)	36 (22.5%)	160

**Table 3 pathogens-08-00242-t003:** Statistical factors associated with the comparison between the indirect immunofluorescence antibody test (IFA) and the CBU_1910 ELISA for the detection of serum phase I IgG antibodies against *Coxiella burnetii*.

TPR	TNR	PPV	NPV	FPR	FNR	lr+	lr−	OR	Acc	95% CI	PE	κ
0.929	0.924	0.722	0.984	0.076	0.071	12.257	0.077	158.6	0.925	32.8–767	0.679	0.767

The statistical analysis was performed based on the ROC. TPR: true positive rate = sensitivity = probability of detection = power = tpr = tp/(tp + fn). TNR: true negative rate = specificity = selectivity = true negative rate = tnr = tn/(tn + fp). PPV: positive predictive value = 1 − specificity = precision = confidence = tp/(tp + fp). NPV: negative predictive value = tn/(tn + fn). FPR: false positive rate = probability of false alarm, fpr = fp/(fp + tn). FNR: false negative rate = miss rate = fnr = fn/(fn + tp). lr+: positive likelihood ratio = lr+ = tpr/fpr. lr−: negative likelihood ratio = lr− = fnr/tnr. OR: odds ratio = lr+/lr−. Acc: accuracy = classification rate, acc = (tp + tn)/(tp + tn + fp + fn). 95% CI = 95% confidence interval. PE: the probability of agreement by chance, PE = ((tp + fp) × (tp + fn) + (tn + fp) × (tn + fn))/((tp + tn + fp + fn) × 2). κ: Cohen’s kappa coefficient of agreement, κ = (Acc − PE)/(1 − PE).

**Table 4 pathogens-08-00242-t004:** Area under the curve analysis. Significance was set at *p* > 0.05.

Area under the Curve					
Protein	Area	Standard Error	Asymptotic Significance	Asymptotic 95% Confidence Interval	
				Lower Bound	Upper Bound
CBU_1910	0.974	0.012	0.000	0.951	0.997

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
