# Peer review of "Com1 as a Promising Protein for the Differential Diagnosis of the Two Forms of Q Fever"

_pathogens, 2019, doi:10.3390/pathogens8040242_

Round 1
Reviewer 1 Report
In this manuscript authors develop a methods to improved Q ferver diagnosis
The results support the idea that a Com1 antigen-based serodiagnostic test may be useful for differential diagnosis of chronic Q fever.
my major comments are:
ethical issue: the origin of the human serum used, the authorisation to use this serum.
concerning the human serum: Several information are missing. I have no idea about how many women's, mans, ages, treatment....
Figure 3 is not convincing, need better quality, controls
Reviewer 2 Report
Major comments
This manuscript has contradictory statements as to the aim of this study. To illustrate, the title indicates that a protein, “Com1 is a Promising Protein for the Differential Diagnosis of Q Fever” (Lines 2-3), which both suggests that (1) the major finding of this study is the application of com1 (in an indirect ELISA) to detect patients with Q fever (infection with Coxiella burnetii), suggesting it is useful for individuals with any C. burnetii infection—and (2) that this assay is useful to detect cases of C. burnetii infection among cases with other clinically similar diseases, i.e. causing acute undifferentiated febrile illness. As for differential diagnosis, there is no real attempt made to differentiate C. burnetii infection from other diseases. Then at the end of the introduction, the authors state that they aimed to “differentially diagnose chronic Q fever in humans.” (Lines 78-79).
First, others have previously purified recombinant com1 and used it as an ELISA antigen to detect human antibody of those it to discriminate Q fever (Zhang et al., 1998) Microbiol. Immunol. 42(6), 423-428. Similar studies were performed subsequently, notably in Marseille (Sekeyova et al., 2010) Acta Virologica and by the US Navy (Chen et al., 2014) Int. J. Bacteriol. Indeed, Sekeyova et al. (2010) noted, “Our results indicated that ELISA using Com1 would be better applicable for the diagnosis of Q fever IE than for Q fever acute cases.” The authors failed to cite Zhang et al. (1998) and build on subsequent studies with an incomplete literature review. It is surprising that they discuss Sekeyova et al. (2010), which directly addresses/answers the aim as stated at the end of the introduction.
As for the utility of com1 to diagnose cases of “chronic” Q fever, the authors need to provide clearer context and additional rationale. It is worth noting that the group in Marseille led by Didier Raoult argues against the use of the term chronic Q fever (Million & Raoult, 2017) Emerg. Infect. Dis. Unfortunately, your introduction does not provide any context for this controversy or clearly explain the pathogenesis of C. burnetii infection in a matter that would make this classification clear.
Minor comments
The methods are missing some important information, such as important details for cloning: “The PCR products were visualized in a 1.5% agarose gel, analyzed by restriction digest and cloned into the destination vector, pET-22b.” (Lines 94-95) is terribly insufficient.
Where you have followed others’ methods, you need to provide citations.
Some methods are included in the results section, e.g. section 3.2.
The gel images do not contain standards for comparison, nor negative or positive controls.
The construct sequence is not provided.
In general, it will be helpful for you to follow the previously mentioned citations for examples.
Reviewer 3 Report
Introduction – explanation pf phase I and phase II antibodies and the likely titres of these in relation to acute or chronic Q fever is required. At the moment why the authors are placing value on a particular titre to phase I as a cut off is not clear to anyone not very familiar with the specific literature. Presumably they are using a particular phase I titre as a proxy measurement for chronic Q fever, but this is not clear.
Methods
line 128-129 – on what basis was the sera considered positive?
Line 138 – was the commercially available kit an IFA?
Line 147 – Q fever positive on what basis?
Results
Figure 1 – doesn’t indicate the restriction sites used but is also unnecessary as it is describing a routine technique using a commercial cloning vector – can be removed (should still be described in methods, which needs the RE sites added) or put into supplementary materials.
Figure 2 – what is the expected apparent molecular weight of Com1 – should be indicated on figure, is the SDS-PAGE Coomassie stained? The immunoblot needs a negative control i.e. E. coli carrying the empty vector.
Fig 3 – given that the authors discuss the difficulty in identifying patients with chronic Q fever with current diagnostic tests how was the “positive”serum selected for testing with Com1 - what diagnostic test was used? Was only one sample tested in immunoblotting or was the sera pooled from a number of patients? Or were a number of serum samples tested separately but a representative result shown? It is not clear from legend or results. Presumably lane 1 is the positive lane – what size should the band be? At the moment only a smear is evident – why is this the case?
The results described in lines 201-216 is very hard to follow and Table1, which I think is a summary, is also unclear as to the significance of the different Q fever titres. Why is 1/12 of the negative controls “positive” – is this likely to be a false positive, or given that Q fever is often asymptomatic, could this be a true positive? This does not seem to be discussed.
Discussion
This is very short and could be combined with results. How do the findings in this study relate t that of Kowakzewska et al 2012 who also looked at many antigens including Com1? What percentage of false positives and negatives is ideal if the candidate antigen is suitable for development into a commercial assay?
Round 2
Reviewer 1 Report
Autors answers to my request
Author Response
We thank the reviewer for giving us the chance to improve our manuscript.
Reviewer 2 Report
1. Com1 should be Italic.
Author Response
We have revised the manuscript and we thank the reviewer for giving us the chance to improve it.
Reviewer 3 Report
The paper has been substantially improved with correct controls and better descriptions of results now included. Some improvement to the level of scientific English and correction of minor grammatical errors is still required.
Author Response

(The authors gave the same response as above.)
